# The Improvement of Performance through Minimizing Scallop Size in MEMS Based Micro Wind Turbine

**DOI:** 10.3390/mi12101261

**Published:** 2021-10-17

**Authors:** Young Chan Choi, June Soo Kim, Soon Yeol Kwon, Seong Ho Kong

**Affiliations:** School of Electronic and Electrical Engineering, Kyungpook National University, Daegu 41566, Korea; chani0988@naver.com (Y.C.C.); junesookim@gmail.com (J.S.K.); yosiki3514@naver.com (S.Y.K.)

**Keywords:** turbine, rotor, blade, MEMS, DRIE, scallop

## Abstract

In this paper we report on the improvement of performance by minimizing scallop size through deep reactive-ion etching (DRIE) of rotors in micro-wind turbines based on micro-electro-mechanical systems (MEMS) technology. The surface profile of an MEMS rotor can be controlled by modifying the scallop size of the DRIE surface through changing the process recipe. The fabrication of a planar disk-type MEMS rotor through the MEMS fabrication process was carried out, and for the comparison of the improvements in the performance of each rotor, RPM testing and open circuit output voltage experiments of stators and permanent magnets were performed. We found that the smooth etching profile with a minimized scallop size formed using DRIE results in improved rotation properties in MEMS-based wind turbine rotors.

## 1. Introduction

Recently, in the area of energy harvesting, various studies have focused on micro-gas turbines based on micro-electro-mechanical systems (MEMS) technology for micro-power generation devices [1,2,3,4]. There have been many studies on various materials related to micro-gas turbines and their fabrication processes, and many have reported on MEMS-based micro gas turbines using single crystal wafers [5,6].

In general, these MEMS gas turbine studies seem to focus on how to obtain more energy through improving the performance of the components that make up the turbine [7,8,9,10,11,12,13,14,15,16,17].

In particular regard to rotors, which are an important component of turbines, many papers have reported on simulations and experiments to optimize and validate the hydrodynamic geometry of the blade to increase the rotational speed of the rotor [18,19].

In addition, the fabrication technology used to produce the rotor without errors is just as important as the design of the blade geometry of the rotor [20]. Silicon-based MEMS micro-gas turbine rotors are mainly fabricated using two major semiconductor processes, the photolithography process and the deep reactive ion etching (DRIE) process. The pattern design of the photo mask used in the photolithography process defines the geometry of the MEMS rotor blade and is an important part of the hydrodynamic design in the two-dimensional space.

It may be thought that the photolithography process is the most important process in rotor design because the area corresponding to the 2D design of the rotor is defined in the photo process, whereas the DRIE process corresponds to the vertical etching of the blade, which only affects depth without any special features. However, the DRIE process is just as important and must be as carefully considered as the photolithography process, which involves the design of the 2D geometry, because the DRIE process results in scalloping during the fabrication process, determining the surface profile. Depending on the DRIE recipe conditions, the fabricated result of the DRIE process has a bumpy surface in the form of the scallops in micro-units, rather than an ideal vertical plane. Therefore, if the DRIE process conditions are changed, different scallop sizes will be formed, which will cause differences in the surface profile of the rotor. In other words, even if two rotors have been fabricated using the same photo mask, the etching surfaces formed through the DRIE process may exhibit different etching profiles depending on the DRIE recipe conditions, which may result in differences in the performance of each rotor.

Surface roughness in hydrodynamics has been reported as an important factor affecting the performance of some devices [21,22]. Therefore, in this paper, we analyzed the shape of the scallops that occur when MEMS turbine rotor blades are fabricated through the DRIE process and studied the DRIE process recipe conditions in order to optimize the performance of the MEMS turbine rotor.

## 2. Materials and Methods

The shape of the MEMS rotor blade used in this study is shown in Figure 1. As in previously reported MEMS gas turbines, the proposed MEMS turbine rotor was designed to allow air to enter vertically from the top of the center of the rotor, flow through the sidewall of the rotor blade, and escape in the horizontal direction of the rotor. The airflow in the planar disk-type rotor causes the rotor to rotate as it passes through the sidewall of the blade shape, which is defined through the photo lithography process.

This flow of air is closely related to the surface conditions of the blade, as it follows the surface of the blade, and Bammert, K. and Sandstede, H. have reported from experiments that an increase in the surface roughness of the turbine rotor blade leads to an increase in the air friction coefficient [23]. Previous studies have demonstrated performance changes in bulk-sized rotor blades with roughness damage acquired due to pollution caused by the environment in which they are used [24,25].

On the other hand, we aimed to study the performance changes caused by the innate sidewall roughness of a 10 mm small planar disk-type MEMS rotor blade determined through the DRIE process as shown in Figure 1. When a planar disk-type MEMS rotor blade is fabricated using the DRIE process, a scallop is formed on the sidewall of the rotor blade as shown in Figure 1b,c, which determines the surface profile of the rotor blade. For a more detailed explanation, the fabrication process of the MEMS turbine rotor is shown in Figure 2.

The proposed MEMS rotor in this study was fabricated using a double-sided polished silicon wafer with a thickness of 680 µm. The high of rotor blade was 380 µm and the rotor disc supporting the blade was 300 µm thick, which is quite thick. A thick masking layer would be needed to fabricate the proposed rotor through the DRIE process. Therefore, in this paper, the thermal oxide layer and the photoresist layer were configured into two layers and used as a thick masking layer for DRIE.

The proposed rotor requires a two-sided DRIE process, which complicates the manufacturing process. Therefore, to minimize the process steps, the oxides on the front side and back side were etched at the same time using wet chemical etching. The fabrication process of the proposed MEMS turbine rotor is illustrated in Figure 2. First, a 1-µm-thick silicon dioxide is grown on both sides of double-side polished silicon wafer through wet oxidation, as shown in Figure 2b. Then, the area to be subjected to DRIE is patterned and defined through the photolithography process. Subsequently, a 7-µm-thick photoresist is spin-coated on the front side of the wafer and baked for the removal of the solvent and curing. After the baking of the front side, the baking process after the photoresist coating of the back side is carried out using a convection oven, not a hot plate, with a wafer carrier to prevent it from sticking to the equipment. As shown in Figure 2f, for the photoresist-coated wafer, the pattern is defined first on the front side through the photolithography process, and then the pattern is formed on the opposite side using backside alignment equipment. Afterwards, the wet etching process was performed with the BOE solution to remove the oxide on the front and back of the wafer at the same time. In the DRIE process, the back side of the silicon substrate is etched to a depth of 300 µm following the photoresist pattern, and the front side of the wafer is also etched to a depth of 380 µm by attaching a passivation film on the back-side substrate to prevent helium leaking during the DIRE process, as shown in Figure 2i. After the DRIE process, we removed the leak passivation film and released the fabricated rotor from the wafer. Finally, we stripped the remaining photoresist and thermal oxide from the surface of the rotor. Figure 3 shows the completed MEMS rotor.

The aforementioned DRIE process is a vertical and anisotropic etching technique invented in the 1990s by Robert Bosch GmbH, called the "Bosch Process". This technology enables silicon etching with a high aspect ratio, which can be used to manufacture MEMS turbine rotors. Figure 4 shows the concept of the Bosch process and the scalloping that occurs in the Bosch process. The Bosh process involves a repeating sequence of etching and passivation processes. These iterative processing methods necessarily result in the etching surface having a ribbed microstructure called a scallop and having different sizes depending on the process conditions. In this paper, to distinguish between the different sizes of these scallops, we defined it by measuring the vertical etching depth (D) and lateral etching width (W), as shown in Figure 4. 

Table 1 provides the DRIE process condition recipes for fabricating different sizes of scallops. In this experiment, the main method used to reduce the size of the scallop was the changing of cycle time and ratio. Reducing the deposition and etching time naturally reduces the vertical etching depth per cycle, making the scallop smaller. Recipe 2 was set up for the purpose of decreasing the scallop size by reducing the cycle time compared to recipe 1, and the ratio of deposition and etch cycle time remained the same as in recipe 1, at 0.57. Recipe 3 was set to increase the ratio of the etching and deposition time to 1.25 by reducing the etching time to obtain a smaller scale size than that in recipe 2. However, if only the etching time is reduced, the etch rate becomes too low to be applied to a 380-µm-deep silicon etching. Therefore, the etching rate in recipe 3 was improved by lowering the flow rate of C_4_F_8_ in the deposition cycle and increasing the O_2_ ratio in the etching cycle [26,27,28].

## 3. Results and Discussions

Figure 5 presents SEM images of the DRIE scallop cross-sections fabricated according to conditions shown in Table 1. The vertical etching depths of the scallops were 0.073, 0.37, and 1.75 µm, respectively. In other words, even if the same design is used for the photo mask pattern, different scallops are formed according to the DRIE recipe conditions, and therefore rotors with different etching profiles can be obtained. The measurement system was configured as shown in Figure 6 to test the characteristics of the fabricated MEMS rotors.

An air compressor was used to supply air to rotate the rotor, and an air regulator was used to control the supplied air flow. The nozzle was located at the top of the rotor so that air was supplied from the direction perpendicular to the rotor. In addition, the nozzle was installed on the xyz stage for precise distance adjustments with the rotor, allowing repeated experiments for different scallop sizes with the same gap. For the rotation test, the fabricated rotor was directly attached to the shaft and connected to the ball bearing, and then the rotor speed was measured using an infrared tachometer. In addition, the open circuit output voltage characteristics of the fabricated rotor combined with permanent magnets and a stator were measured to verify the power generation properties caused by rotation.

Figure 7 presents a schematic diagram of the components of the turbine constructed for the open circuit output voltage experiment of the fabricated rotor. A four-pole permanent magnet in the form of a disk was bonded to the back side of the fabricated rotor, and the shaft was attached to the rotor and mounted on the ball bearing, as shown in Figure 7. In addition, a planar coil stator made for power induction through permanent magnet rotation was positioned facing the permanent magnet.

Figure 8 presents the principle diagram of an induced current experiment using the planar stator. When a donut-shaped permanent magnet rotates, an electric current is induced on a fixed stator. The proposed stator is not only flat but has three phases.

Figure 9 shows the measured results of rotor speed according to scallop size using the configured system.

As the vertical etching depth of the scallop decreases, the number of rotations of the rotor increases. The change in the number of rotations due to the effect of scallop size was similar at a low pressure, but as the pressure increased, it became larger. As the size of the scallop became smaller, it became closer to the smooth surface and we observed that the performance improved. Furthermore, an improved number of rotations was observed in inverse proportion to the scallop changes in the experiment, so it was determined that the surface without scalloping would exhibit the best rotations.

The measurement results of the open circuit output voltage system using the fabricated rotor are shown in Figure 10.

For the experiment, the same air pressure (20 psi) was supplied to the rotor and the generation characteristics were observed according to the scallop size. The results showed that the frequency of the output voltage increased because the rotary motion of the rotor increased as the scallop size decreased. Furthermore, the higher the frequency of *V_rms_*, the higher the power of *V_rms_*, which was judged to be a natural phenomenon of higher generation characteristics caused by higher rotational rates. In other words, the results of this experiment showed that a smooth surface caused by small scallop size increases the number of rotations of the rotor, resulting in an increased open circuit output voltage due to the increased number of rotations.

## 4. Discussions

In this paper, we report on the improvement of the rotation speed and open circuit output voltage of membrane rotors by minimizing the scallop size through to the optimization of DRIE process conditions. It is necessary to carefully discuss whether the experimental results of this paper were influenced only by the size of the scallop. This is because the size of the scallop changes according to the DRIE recipe, but the volume of the blade etched through DRIE will also change, causing changes in the mass of the rotor. The difference in the theoretically calculated mass of the rotors in this paper was 0.000024 g, which was too small to be measured using general-scale equipment. Therefore, it should be considered that the results shown in this paper include a fine difference in rotor mass and blade volume, related to the torque of the rotor, which are difficult to measure. However, the main topic of this paper is that even if the same photo mask pattern is used, the performance of the rotor blade is affected by the DRIE process conditions, and for performance optimization, the DRIE process conditions must be optimized when manufacturing MEMS rotors.

## 5. Conclusions

In this paper, we report on performance improvements through minimizing the size of the sidewall scallops produced by the DRIE process in fabricating the rotors of micro-wind turbine based on MEMS technology. Silicon wafers with the same photo patterns created through the photolithography process with the same mask were etched under different DRIE process conditions. The sizes of the formed sidewall scallops were 0.073, 0.37, and 1.75 µm, which can be represented by the etching profiles of the blades, respectively.

To analyze the performance of the proposed rotors with micro-scale blades, the rotor speed and the open circuit output voltage characteristics according to the scallop size were compared. The experimental results indicate that the rotational speed of the rotor decreases as the scallop size of the blade surface increases under the same air pressure conditions. The open circuit output voltage performance of scallop sizes 1.75 and 0.073 µm obtained in this paper differed by up to 13%. Thus, it was empirically confirmed that it is important to optimize the etching profile by adjusting the scallop size.

Considering these results, when manufacturing MEMS-based blades, controlling the DRIE conditions can be advantageous for the rotational performance of the micro-wind turbine. Through this application, it is expected that the research field of MEMS rotary generators with improved performance will be developed.

## Figures and Tables

**Figure 1 micromachines-12-01261-f001:**
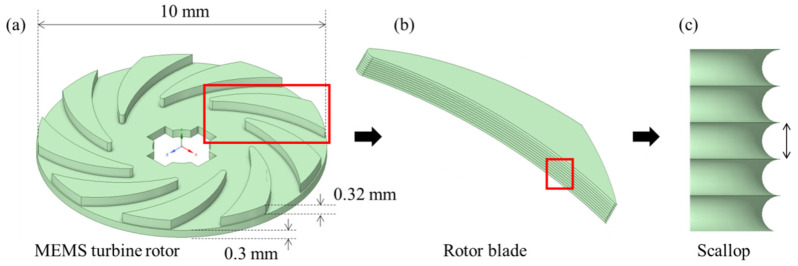
(**a**) Proposed MEMS turbine rotor, (**b**) rotor blade, and (**c**) sidewall scallop of the rotor blade.

**Figure 2 micromachines-12-01261-f002:**
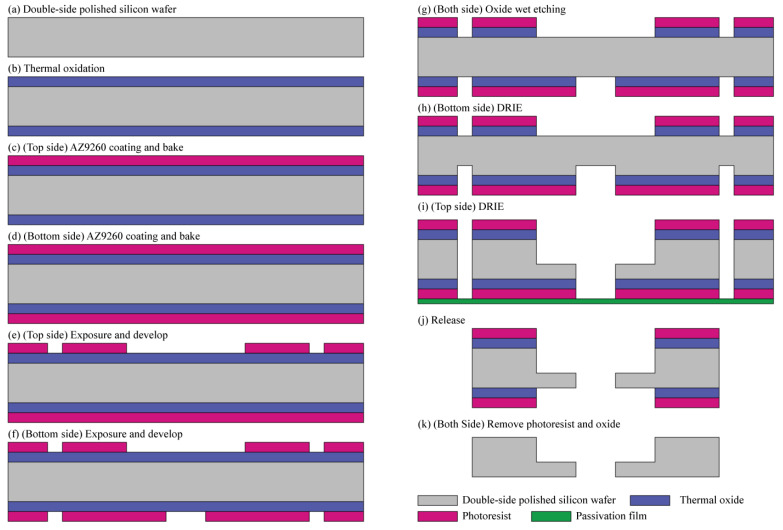
Fabrication process flow of the proposed MEMS turbine rotor.

**Figure 3 micromachines-12-01261-f003:**
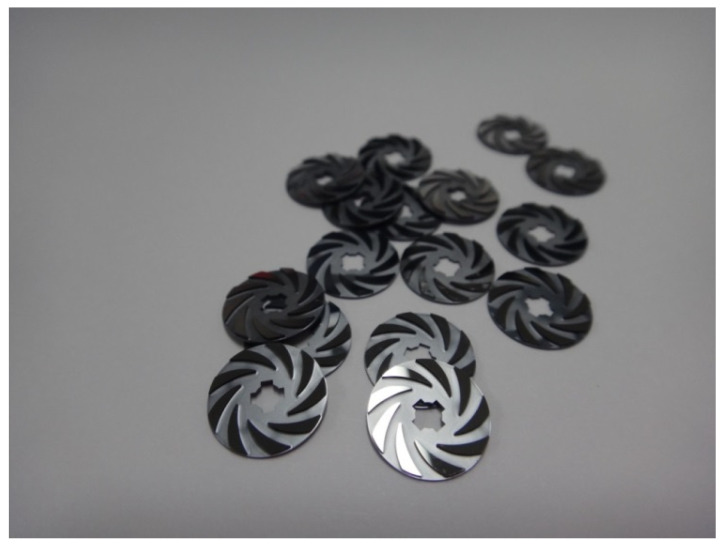
Fabricated MEMS turbine rotor.

**Figure 4 micromachines-12-01261-f004:**
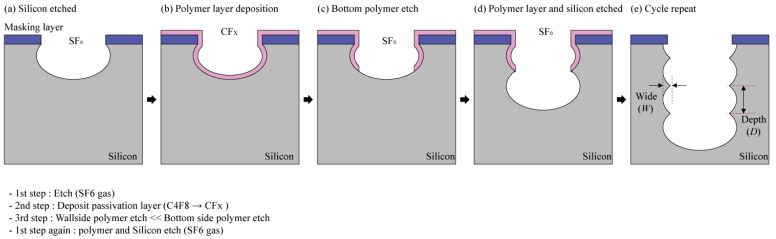
Schematic picture of steps in the Bosch process and the scallops formed.

**Figure 5 micromachines-12-01261-f005:**
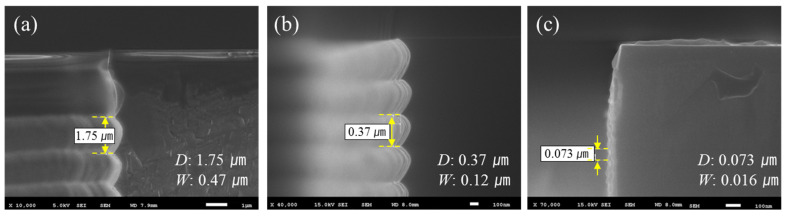
Cross sectional SEM image of the fabricated DRIE scallop. (**a**) Recipe 1, (**b**) recipe 2, and (**c**) recipe 3.

**Figure 6 micromachines-12-01261-f006:**
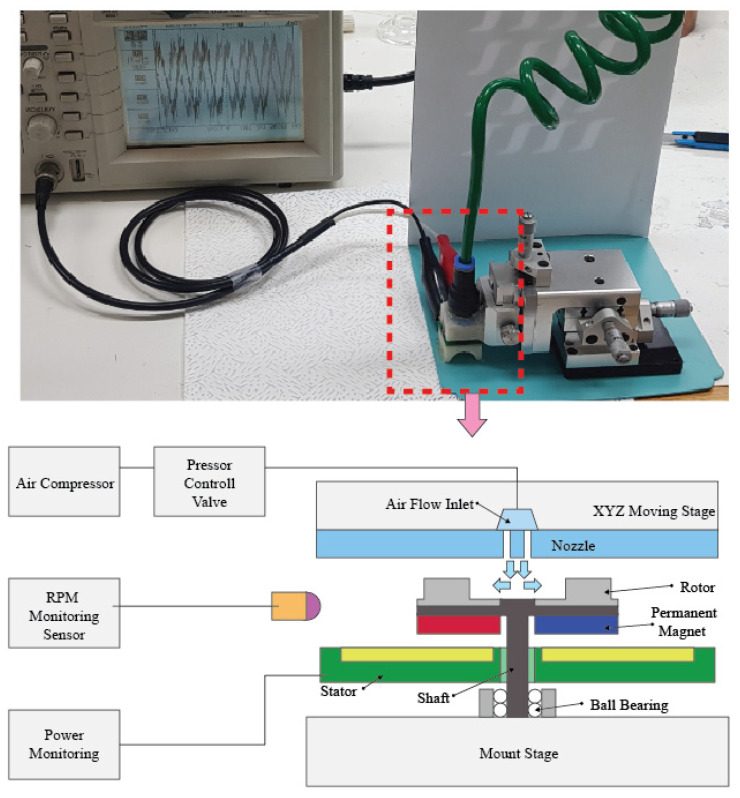
Measurement system for rotation and power-generating performance.

**Figure 7 micromachines-12-01261-f007:**
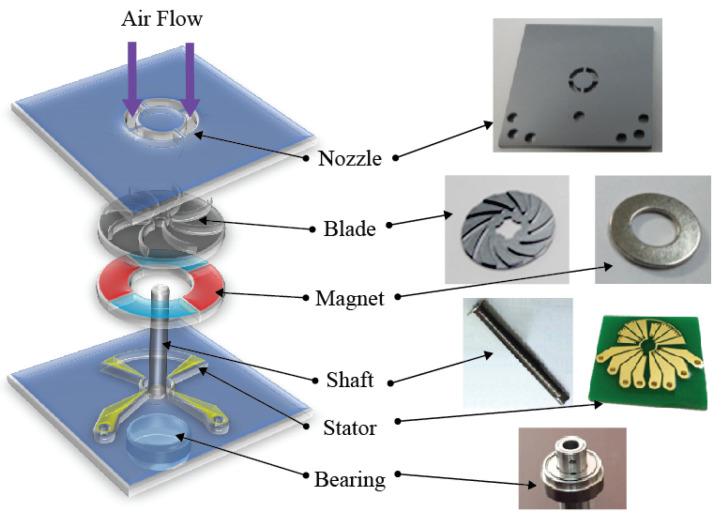
Part assembly diagram for open circuit output voltage experiment of the proposed rotor.

**Figure 8 micromachines-12-01261-f008:**
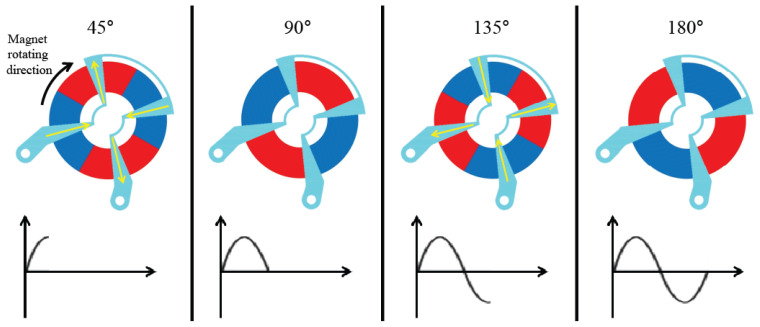
Principle diagram of alternating voltage induced in a planar magnet and coil.

**Figure 9 micromachines-12-01261-f009:**
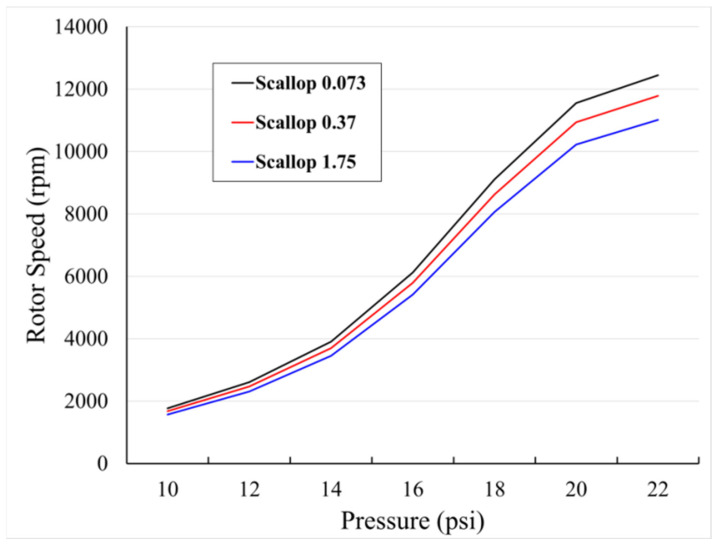
Measured rotor speed curve according to the different scallop size.

**Figure 10 micromachines-12-01261-f010:**
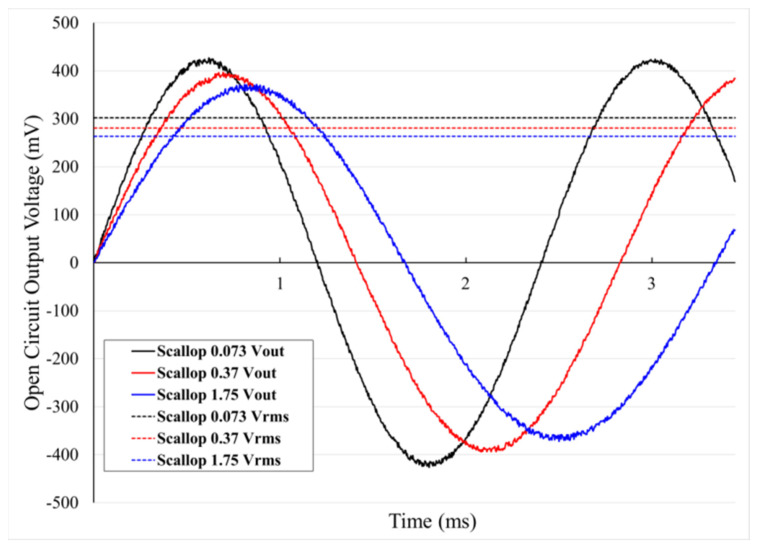
Measurement results of open circuit output voltage according to different sizes of the scallops (pressure = 20 psi).

**Table 1 micromachines-12-01261-t001:** DRIE process condition recipes for different vertical etching depths.

Heading	Cycle	Time (Sec)	Pressure (mTorr)	GAS (sccm)			RF			Chuck Temperature (°C)	Etch Rate (µm/min)
				C_4_F_8_	SF_6_	O_2_	Coil (13.56 MHz)	HF (13/56 MHz)	LF (380 kHz)		
Recipe 1 (Base)	Depo	4	35	270	-	-	2000	-	20	10	8.8
	Etch	7	60	-	400	1	2200	-	25		
Recipe 2	Depo	1.5	35	270	-	-	2000	-	20	10	4.75
	Etch	2.6	60	-	400	1	2200	-	25		
Recipe 3	Depo	1.5	30	200	-	-	2000	-	-	5	3.15
	Etch	1.2	15	30	350	36	2000	60	-		

## Data Availability

The data that support the findings of this study are available from the first author upon reasonable request.

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
