# Peer review of "The Improvement of Performance through Minimizing Scallop Size in MEMS Based Micro Wind Turbine"

_micromachines, 2021, doi:10.3390/mi12101261_

Round 1

Reviewer 1 Report

In this paper, authors report their study on the DRIE recipe to reduce scalloping effect for improving the performance of a micro wind turbine. The concept is simple and straightforward, and this paper can provide new information to audience in this field. However, there are a couple issues need to be addressed before publication.

  1. I would suggest authors put all the background references to the Introduction section, such as reference 23 and Bosch process. A reference paper should be cited for Bosch process.
  2. The dimension of the MEMS turbine rotor should be provided in figure 1. Also, please use consistent wordings, for example, rotor disc or rotor, rotor blade or blade.
  3. For the fabrication process shown in figure2. Please add figure sub-numbers in the text so that readers can follow your description better. Also, it should be double-sided polished silicon wafer in the figure2(1). Each colored layer should also be defined in the figure. Lastly, what was the thickness of the photoresist?
  4. Table 1 summarizes the three DRIE processes compared in this paper. The etching rate should be provided for each recipe. I suspected that the etch rate could be very slow. In this case, the practical application of this process may be limited. Please leverage the potential application for this process.
  5. Authors state that the size of the vertical etching depth of the etched surface can represent surface roughness. This is not a correct approach to define surface roughness. Please use image method to measure and quantify surface roughness for each side wall. A more engineering quantification should use.
  6. Finally, authors state that this micro wind turbine is for the application of power generation. However, the experimental results only provide rotating speed and open circuit output voltage. These results are not sufficient to demonstrate its performance on power generation. A more detailed comparison on electric power generation should provide. Its correlation to the rotation speed should also discuss and compare.

Author Response

Response to Reviewer 1 Comments

Point 1: I would suggest authors put all the background references to the Introduction section, such as reference 23 and Bosch process. A reference paper should be cited for Bosch process.

Response 1: I tried to modify it as much as possible by referring to the reviewer's opinion, but it was very difficult to make a smooth configuration. The author of this paper moved the background references to the introduction session, but it was very difficult to organize the text because the description of scallop was required in the text. I ask for your generous understanding in maintaining this configuration.

Point 2: The dimension of the MEMS turbine rotor should be provided in figure 1. Also, please use consistent wordings, for example, rotor disc or rotor, rotor blade or blade.

Response 2: According to the reviewer's good advice, the dimension of the MEMS turbine rotor was added on Figure 1. And each name of the parts was unified and entered.

Point 3: For the fabrication process shown in figure 2. Please add figure sub-numbers in the text so that readers can follow your description better. Also, it should be double-sided polished silicon wafer in the figure 2(1). Each colored layer should also be defined in the figure. Lastly, what was the thickness of the photoresist?

Response 3: Figure 2 and related descriptions was revised.

  • Sub-number was revised and the color of each layer was defined in Figure 2.
  • Figure 2 (a) Double polished silicon wafer → Double-side Polished silicon wafer
  • (Description) The Sub-number was added (Line 94, 99, 106).
  • (Description) About the thickness of the photoresist (Line 95):
  • Before) Subsequently, a thick photoresist is spin-coated on front side of wafer and baked for removing solvent and curing.
  • After revision) Subsequently, a 7-µm-thick photoresist is spin-coated on front side of wafer and baked for removing solvent and curing.

Point 4: Table 1 summarizes the three DRIE processes compared in this paper. The etching rate should be provided for each recipe. I suspected that the etch rate could be very slow. In this case, the practical application of this process may be limited. Please leverage the potential application for this process.

Response 4: According to reviewer’s advice, the etch rate for each recipe was added to Table 1.

Point 5: Authors state that the size of the vertical etching depth of the etched surface can represent surface roughness. This is not a correct approach to define surface roughness. Please use image method to measure and quantify surface roughness for each side wall. A more engineering quantification should use.

Response 5: Please understand that we could not use quantitative and general surface roughness measurement methods such as AFM (Atomic Force Microscopy). It was difficult to pre-process and prepare specimens for the measurement of the rotor manufactured in a three-dimensional structure, so only vertical cross-sectional analysis of the SEM image was conducted. However, as you all know, DRIE's scallop is quite uniform and repetitive. So, we carefully expect that the level of surface roughness will be sufficiently transmitted through vertical etching depth D.

And according to opinions of other researchers, the term “surface roughness” was replaced and changed to “Etch profile”, which is normally used in the DRIE process filed.

However, according to your useful advice, we added lateral etching wide W by referring to other DRIE research for making more engineering quantification. [Ref 1: Kim, K.; Lee, J.; Han, S.; Lee, S. A Novel Top-Down Fabrication Process for Vertically-Stacked Silicon-Nanowire Array. Appl. Sci. 2020, 10, 1146.]

- Before) In this paper, to distinguish the surface roughness formed according to the profile of the scallop, we define it as a vertical etching depth, as shown in Figure 4.

- After revision) In this paper, to distinguish the surface roughness formed according to the profile of the scallop, we define it as a vertical etching depth D and lateral etching wide W, as shown in Figure 4.

The lateral etched wide concept was added to the figure and modified.

Point 6: Finally, authors state that this micro wind turbine is for the application of power generation. However, the experimental results only provide rotating speed and open circuit output voltage. These results are not sufficient to demonstrate its performance on power generation. A more detailed comparison on electric power generation should provide. Its correlation to the rotation speed should also discuss and compare.

Response 6: Revision completed (Line 15, 152, 158, 164, 181, 193, 196, 218, 221).

Reviewer 2 Report

This paper illustrates the improved power of micro wind turbine through reducing the DRIE caused scallop. The manuscript is well structured and presented. The contents and results are comprehensive and attractive to the researchers in the community. I have the following comments listed for improving the readability of this article:

  1. In the main texts, abbreviations have to be defined before use: Line 22 MEMS, and Line 35: DRIE.
  2. Ref [23] cited in Line 71 is the major motivation of this study. However, it is very out of date (published in 1980). The authors have to include and discuss the more recent literature.
  3. Subsets in Figure 2 are better been marked with (a), (b), (c)… and so on, for each step in the process flow. So that the descriptions of Fig. 2 in the main texts (Lines 89 to 106) can easily refer to a specific step where applicable.
  4. Results in Figure 10 could be from a fixed air pressure, but that is not clearly stated. Also, what are the torque values of each of the three turbine blades with different sizes of scallop?
  5. From Figure 9, a same rotor speed can be obtained when the applied pressure is low (say </= 10 psi). According to the SEM images in Figure 5, there should be some difference in mass between the three turbines due to the difference of scallop depth. The effects of torque, mass, and roughness on the rotor speed are not clear to me.
  6. The term “roughness” in the title cannot unambiguously connect to the DRIE scallop in the context. Roughness is often quantified through Ra, Rq and Rz. Justifications of the effects of roughness on the rotor speed are insufficient as pointed in the above #5 comment.

Author Response

Response to Reviewer 2 Comments

Point 1: In the main texts, abbreviations have to be defined before use: Line 22 MEMS, and Line 35: DRIE. 

Response 1: The Line 22 MEMS and Line 35 DRIE have been revised according to the Reviewer's comment.

Point 2: Ref [23] cited in Line 71 is the major motivation of this study. However, it is very out of date (published in 1980). The authors have to include and discuss the more recent literature.

Response 2: The reference has been changed to other recently reported papers (Line 71).

Point 3: Subsets in Figure 2 are better been marked with (a), (b), (c)… and so on, for each step in the process flow. So that the descriptions of Fig. 2 in the main texts (Lines 89 to 106) can easily refer to a specific step where applicable.

Response 3: Figure 2 and related descriptions was revised in compliance with the reviewer's comments (Line 94, 99, 106).

Point 4: Results in Figure 10 could be from a fixed air pressure, but that is not clearly stated. Also, what are the torque values of each of the three turbine blades with different sizes of scallop?

Response 4: Figure 10 caption (Line 85) and related description (Line 86) was revised to add information about the input air pressure. (20 psi)

Unfortunately, the author is a MEMS process engineer who does not major in mechanical engineering in turbines and does not have the ability to calculate or simulate the torque of the blade, which is causing considerable difficulties in the reviewer's questions. We ask for your appropriate understanding of this situation.

Point 5: From Figure 9, a same rotor speed can be obtained when the applied pressure is low (say </= 10 psi). According to the SEM images in Figure 5, there should be some difference in mass between the three turbines due to the difference of scallop depth. The effects of torque, mass, and roughness on the rotor speed are not clear to me.

Response 5: Referring to what the reviewer pointed out, I added the sentence below.

(Line 175) The change in the number of rotations due to the effect of scallop size was similar at the low pressure, but as the pressure increased, it became larger.

Mass difference between the three turbines was so small that it could not be measured by scale. Based on the density of single crystal silicon wafers, the theoretically expected weight difference between is about 0.000024g. The measurement results could not be presented because there was no equipment capable of accurately measuring small weight differences.

However, based on reviewer’s important comment, I mentioned the possibility that the weight difference of the rotor by DRIE could have affected rotor speed by adding the discussion part (Line 195) as below. And all of the effective factors on rotor speed were caused by DRIE, so the important topic of this paper is DRIE process should be optimized when manufacturing the MEMS rotor. I cautiously suggest that this will be an important issue for researchers designing or manufacturing MEMS rotors.

Point 6: The term “roughness” in the title cannot unambiguously connect to the DRIE scallop in the context. Roughness is often quantified through Ra, Rq and Rz. Justifications of the effects of roughness on the rotor speed are insufficient as pointed in the above #5 comment.

Response 6: In full agreement with the reviewer's comment, the term “surface roughness” was replaced and changed to “etch profile”, which is normally used in the DRIE process filed.

Please understand that we could not use quantitative and general surface roughness measurement methods such as AFM (Atomic Force Microscopy). It was difficult to pre-process and prepare specimens for the measurement of the rotor manufactured in a three-dimensional structure, so only vertical cross-sectional analysis of the SEM image was conducted. However, as you all know, DRIE's scallop is quite uniform and repetitive. So, we carefully expect that the level of surface roughness will be sufficiently transmitted through vertical etching depth D.

And I cautiously suggest that expressing that way will be a more practical way of expression for engineers in charge of the DRIE process who will carefully read this paper.

Round 2

Reviewer 1 Report

In the original manuscript, I raised a couple questions about fabrication process and experimental studies. Authors have replied with an acceptable revision. I think this paper can be accepted for publication after the following question been properly addressed.

  1. Authors provided DRIE etch rate as I suggested. However, I think authors should discuss the difference of these DRIE recipes that makes the recipe 3 can have such a small scallop profile. Since the DRIE process is the major contribution of this paper, I think it is important to discuss the difference of these recipes. Particularly for recipes 2 and 3, the etch rate of recipe 2 only is about 50% larger than recipe 3. But, the scallop size of recipe 3 is 5 times smaller than recipe 2. Was it due to the compositions of gas injected or the RF signal been applied?
  2. There is some problem with reference 24.

Author Response

Point 1: Authors provided DRIE etch rate as I suggested. However, I think authors should discuss the difference of these DRIE recipes that makes the recipe 3 can have such a small scallop profile. Since the DRIE process is the major contribution of this paper, I think it is important to discuss the difference of these recipes. Particularly for recipes 2 and 3, the etch rate of recipe 2 only is about 50% larger than recipe 3. But, the scallop size of recipe 3 is 5 times smaller than recipe 2. Was it due to the compositions of gas injected or the RF signal been applied?

Response 1: Revision completed (main text and reference).

(revision) In this experiment, the main method to reduce the size of the scallop is the change of cycle time and ratio. Reducing the deposition and etch time naturally reduces the vertical etching depth of per cycle, making the scallop smaller. Recipe 2 was set up for the purpose of decrease the scallop size by reducing the cycle time compared to recipe 1, and at this time, the ratio of deposition and etch cycle time remained the same as 0.57 in recipe 1. Recipe 3 is set to increase the ratio of etch and deposition time to 1.25 by reducing etch time to obtain a smaller scale size than recipe 2. However, if only the etch time is reduced, the etch rate becomes too low to be applied to a 380 µm depth silicon etching. Therefore, the etch rate of the recipe 3 was improved by lowering the flow rate of C4F8 in the deposition cycle and increasing the O2 ratio in the etch cycle [26, 27, 28].

Point 2: There is some problem with reference 24.

Response 2: Revision completed.
